

# Metabolomics combined with clinical analysis explores metabolic changes and potential serum metabolite biomarkers of antineutrophil cytoplasmic antibody-associated vasculitis with renal impairment

Siyang Liu[1], Qing Xu[1], Yiru Wang[1], Yongman Lv[1,2] and Qing quan Liu[1]

[1] Department of Nephrology, Tongji Hospital, Tongji Medical College, Huazhong University of Science and Technology, Wuhan, Hubei, China
[2] Department of Health Management Centre, Tongji Hospital, Tongji Medical College, Huazhong University of Science and Technology, Wuhan, Hubei, China

Corresponding author
Qing quan Liu, qqliutj@163.com

## ABSTRACT

**Background.** Antineutrophil cytoplasmic antibody (ANCA)-associated vasculitis (AAV) is an autoimmune systemic disease, and the majority of AAV patients have renal involvement presenting as rapid progressive glomerulonephritis (GN). Currently, the clinically available AAV markers are limited, and some of the newly reported markers are still in the nascent stage. The particular mechanism of the level changes of various markers and their association with the pathogenesis of AAV are not well defined. With the help of metabolomics analysis, this study aims to explore metabolic changes in AAV patients with renal involvement and lay the foundation for the discovery of novel biomarkers for AAV-related kidney damage.

**Methods.** We performed liquid chromatography-tandem mass spectrometry (LC-MS/MS)-based on serum samples from patients with AAV ($N = 33$) and healthy controls ($N = 33$) in order to characterize the serum metabolic profiling. The principal component analysis (PCA) and orthogonal partial least-squares-discriminant analysis (OPLS-DA) were used to identify the differential metabolites. Least Absolute Shrinkage and Selection Operator (LASSO) and eXtreme Gradient Boosting (XGBoost) analysis were further conducted to identify the potential diagnostic biomarker. A receiver operating characteristic (ROC) curve analysis was conducted to evaluate the diagnostic performance of the identified potential biomarker.

**Results.** A total of 455 metabolites were detected by LC-MS analysis. PCA and OPLS-DA demonstrated a significant difference between AAV patients with renal involvement and healthy controls, and 135 differentially expressed metabolites were selected, with 121 upregulated and 14 downregulated. Ninety-two metabolic pathways were annotated and enriched based on the KEGG database. N-acetyl-L-leucine, Acetyl-DL-Valine, 5-hydroxyindole-3-acetic acid, and the combination of 1-methylhistidine and Asp-phe could accurately distinguish AAV patients with renal involvement from healthy controls. And 1-methylhistidine was found to be significantly associated with the progression and prognosis of AAV with renal impairment. Amino acid metabolism exhibits significant alternations in AAV with renal involvement.

**Conclusion**. This study identified metabolomic differences between AAV patients with renal involvement and non-AAV individuals. Metabolites that could accurately distinguish patients with AAV renal impairment from healthy controls in this study, and metabolites that were significantly associated with disease progression and prognosis were screened out. Overall, this study provides information on changes in metabolites and metabolic pathways for future studies of AAV-related kidney damage and lays a foundation for the exploration of new biomarkers of AAV-related kidney damage.

## INTRODUCTION

Antineutrophil cytoplasmic antibody (ANCA)-associated vasculitis (AAV) is an autoimmune systemic disease that can affect organs such as the kidneys, heart, lungs, and digestive tract, characterized by an inflammatory reaction in the wall of small vessels and fibrinoid necrosis seen in the pathological tissue, mainly invading small vessels (*Brogan & Eleftheriou, 2018*; *Nakazawa et al., 2019*). AAV can be divided into granulomatosis with polyangiitis (GPA), microscopic polyangiitis (MPA), and eosinophilic GPA (EGPA) based on clinical phenotype, or PR3-ANCA disease *vs* MPO-ANCA disease based on ANCA specificity. The kidney is the most affected organ in AAV. Studies have shown that over 75% of AAV patients have renal involvement, which is one of the leading causes of mortality in AAV patients, and its prognosis is closely related to the renal function of the patient at the time of diagnosis (*Geetha & Jefferson, 2020*; *Sinico, Di Toma & Radice, 2013*). The typical renal presentation is rapidly progressive glomerulonephritis with decreased renal function accompanied by proteinuria, microscopic hematuria, and hypertension for days to months. Patients are often diagnosed in the stage of renal failure due to the insidious onset and rapid progression (*Binda, Moroni & Messa, 2018*). Therefore, biomarkers for AAV diagnosis, progression monitoring, and prognosis predicting for AAV with renal involvement are urgently needed to improve patients' therapeutic outcomes.

In recent years, the classical marker ANCA has been studied intensively. ANCA is currently used as the most important biomarker for AAV diagnosis. However, up to 10% of patients with AAV still test clinically negative for ANCA. In contrast, false positive results are found in the general population and are associated with infections, malignancies, and autoimmune gastrointestinal and kidney diseases (*Bossuyt et al., 2017*; *Houben et al., 2016*; *McAdoo et al., 2012*). The value of ANCA in monitoring progression and predicting relapse is also controversial. Some studies indicate that ANCA may have a role in predicting the recurrence of AAV in individuals with renal or pulmonary involvement, although its function in predicting granulomatous disease is limited (*Fussner et al., 2016*; *Kemna et al., 2015*). But there were also quite a few studies that support the view that ANCA is weakly correlated with disease activity (*Finkielman et al., 2007*; *Tomasson et al., 2012*).

Many new biomarkers for the diagnosis, progression detection, and prognostic analysis of AAV have been reported successively. Recent studies have shown that the activation of the

complement bypass pathway plays an important role in the pathogenesis of AAV, and some members of the complement system such as C3a, C5a, soluble C5b-9, Bb and complement factor H can also act as biomarkers (*Chen et al., 2015*; *Gou et al., 2013a*; *Gou et al., 2013b*; *Wu et al., 2019*; *Yuan et al., 2012*). LAMP-2 is a promising biomarker with a detection rate of up to 90% in untreated AAV patients and is frequently undetectable in the absence of clinical disease activity, indicating that it is associated with disease activity. However, there is no feasible detection method for LAMP-2 for clinical application, which limits its further verification and application (*Kain et al., 2008*; *Kain et al., 1995*; *Kain et al., 2012*; *Peschel et al., 2014*). Similarly, anti-PLG antibodies were found to be elevated in serum in patients with AAV, and anti-PLG levels correlated with disease activity and renal involvement while being limited by the lack of suitable detection methods (*Berden et al., 2010*; *Hao et al., 2014*). Some biomarkers associated with inflammation, including HMGB1, B-cell activating factor (BAFF), soluble urokinase plasminogen activation receptor (suPAR), and urinary biomarkers such as monocyte chemoattractant protein-1(MCP-1), sCD163 and Gremlin are also reported (*Droguett et al., 2019*; *Huang et al., 2020*; *O'Reilly et al., 2016*; *Tam et al., 2004*; *Wang et al., 2013*; *Xin et al., 2014*).

However, currently, the clinically available AAV markers are limited, and some of the newly reported markers are still in the nascent stage. The particular mechanism of the level changes of various markers and their association with the pathogenesis of AAV are not well defined, and a significant number of clinical investigations are still required to verify these findings. Increasing research findings have suggested that metabolic alterations play an important role in autoimmune diseases by providing energy and specific biosynthetic precursors to regulate the growth, differentiation, survival, and activation of immune cells (*Colamatteo et al., 2019*; *O'Neill & Hardie, 2013*; *Stathopoulou, Nikoleri & Bertsias, 2019*). Yet, there have been few studies to date focusing on the metabolic changes in AAV with renal involvement (*Geetha et al., 2022*). The high-throughput, high-resolution phenotyping enabled by metabolomics has been increasingly applied in nephrology research for the analysis of disease mechanisms and promising biomarkers (*Kalim & Rhee, 2017*). We anticipate that the application of the metabolomic technique in AAV with renal involvement will provide us with windows of opportunities to explore promising biomarkers for diagnosis, progression monitoring, and prognosis assessment, and screen intervention sites available for clinical treatment.

## MATERIALS & METHODS

### Clinical samples
This study was approved by the Medical Ethics Committee of Tongji Hospital of Huazhong University of Science and Technology (TJ-IRB20220159). The Medical Ethics Committee granted an exemption from the requirement for informed consent because the serum samples we collected were the samples left over from the participants' routine blood tests, and the study would not affect the rights or health of participants.

Thirty-three patients with AAV in the department of nephrology at Tongji Hospital from June 2015 to July 2017 were recruited in this study, and they were followed up until

December 2019. All of them were newly diagnosed with AAV-related renal impairment and had not received any immunosuppressive therapy prior to sampling. An equal number of healthy controls from the health management center of Tongji Hospital were enrolled in this study. Plasma samples were collected and stored at −80 °C for experimental use. The patients were tested positive for ANCA antibodies by immunofluorescence and enzyme-linked immunosorbent assay, and their clinical diagnoses were confirmed as AAV with renal involvement. All of the patients' symptoms met the criteria of the 2012 Chapel Hill Consensus Conference definition for AAV (*Jennette et al., 2013*). Patients with metabolic syndrome, malignancy, diabetes, hyperthyroidism and hyperlipidemia were excluded because these diseases had great effects on patient's serum metabolic profile, which would have interfered with the results of this study. We also excluded patients with other kidney diseases, other autoimmune diseases and patients taking immunosuppressive drugs. Because they have similar clinical presentation or pathogenesis to AAV patients with renal involvement, there may be a lot of overlap in metabolic changes, which may mask the specific metabolic changes of AAV patients with renal involvement. The healthy controls were selected based on gender matching to eliminate gender differences from the results.

## Sample preparation

Frozen samples were taken out from the −80 °C refrigerator and thawed at 4 °C. Taking 100 μl of each plasma into an EP tube and adding 300 μL of methanol. The mixtures were vortexed for 3 min and then centrifuged at 12,000 r/min for 10 min at 4 °C. The supernatants were finally transferred to the injection bottle for LC-MS/MS analysis. Equal volumes of the separated samples were utilized to generate the pooled plasma sample, which was used to assist quality control, ensure the high-quality of data collected in batches by the high-resolution mass spectrometer, and assess the repeatability of the LC-MS/MS system.

## LC-MS/MS analysis

We adopted broadly targeted metabolome technology to analyze the metabolomes of plasma samples from AAV patients with renal involvement and healthy controls. The data acquisition instrumentation system mainly consisted of Ultra Performance Liquid Chromatography (UPLC) (Shim-pack UFLC SHIMADZU CBM30A; Shimadzu, Kyoto, Japan) and tandem mass spectrometry (MS/MS) (4500 QTRAP; Applied Biosystems, Foster City, CA, USA). An ACQUITY UPLC HSS T3 column (2.1 mm i.d. × 100 mm, 1.8 μm; Waters) was used in UPLC to analyze the metabolomes of interest. And the quantification of metabolites was carried out using the multiple reaction monitoring mode of triple quadrupole mass spectrometry. The samples were placed in an autosampler maintained at 40 °C, and then 2 μl samples were injected for LC-MS/MS analysis.

## Data analysis and visualization

Firstly, the software Analyst 1.6.1 (https://sciex.com/products/software/analyst-software) was used to process mass spectrometry data. The raw data of LC-MS/MS were qualitatively analyzed based on the metware database and the public database of metabolite information and quantitatively analyzed by the software MultiaQuant.

**Table 1** Clinical characteristics of human subjects.

| | Patients (N = 33) | Controls (N = 33) | P values |
|---|---|---|---|
| Male/Female | 16/17 | 16/17 | |
| Age (years) | 49.09 ± 15.84 | 40.21 ± 9.80 | 0.008 |
| BUN (mmol/L) | 16.49 ± 7.88 | 8.04 ± 13.48 | 0.003 |
| Creatinine (mmol/L) | 451.30 ± 316.08 | 75.06 ± 14.94 | <0.001 |
| Total cholesterol (mmol/L) | 4.07 ± 1.31 | 4.12 ± 0.54 | 0.856 |
| eGFR (ml/min/1.73 m$^2$) | 20.81 ± 20.92 | 102.96 ± 21.14 | <0.001 |

Next, the qualitative and quantitative data were analyzed and visualized using PCA, OPLS-DA, volcano plots, and heatmaps to understand metabolic differences between groups and screen for differential expressed metabolites (DEMs). The KEGG database was used to annotate the differential metabolites and identify metabolic pathways associated with them. The analyses mentioned above were achieved with the R Programming Language (*R Core Team, 2021*) (base package; MetaboAnalystR; ComplexHeatmap).

Lastly, subjects were divided into 4 combinations by disease and health, male and female, and each combination was randomly split into training and validation sets in a 2:1 ratio. We applied R (glmnet) and R (xgboost) to perform Least Absolute Shrinkage and Selection Operator (LASSO) and eXtreme Gradient Boosting (XGBoost) in the training set to screen variables and obtain the best combination for diagnosing biomarkers with the use of a regression model. Unpaired $t$-test and receiver operating characteristic (ROC) curves were used to estimate the selected biomarker combination in both the training and validation sets. Data were expressed as mean ± standard deviation (mean ± SD), and $P < 0.05$ were considered statistically significant. GraphPad Prism software (Graph Software, San Diego, CA, USA) was used to generate graphs.

## RESULTS

### Demographic and clinical characteristics of subjects

The clinical characteristics of the study cohort are summarized in Table 1. The two groups have equal numbers of male and female subjects. The AAV group has much higher values of BUN and creatinine (mean 16.44 mmol/L and 476.16 mmol/L respectively) than the healthy group (mean 8.17 mmol/L and 75.90 mmol/L respectively). The eGFR of the AAV group (mean 19.25 ml/min/1.73 m$^2$) significantly declined compared with the healthy group (mean 103.38 ml/min/1.73 m$^2$). There is a statistical difference in age between the two groups. However, given the average ages are similar (48.88 years for the AAV group and 40.25 years for the healthy group), it is anticipated that the difference would not affect the findings or interpretation of the data.

### Metabolome profiling analysis of serum samples obtained from AAV patients and healthy controls

Broadly targeted metabolome technology was used to comparatively analyze the serum samples collected from the AAV patients with renal involvement ($n = 33$) and the healthy

controls ($n = 33$). A total of 455 metabolites were detected and quantified. The curves of the total ion current of the quality control samples were highly overlapping, that is, the retention time and peak intensity were consistent, indicating that the instrument had high stability, which was an important guarantee for the repeatability and reliability of the data (Fig. S1).

Unsupervised PCA preliminarily demonstrated a significant separation between AAV samples and controls PC1 (23.38%) and PC2 (6.3%) (Fig. 1A). Similarly, the two groups were separated in a three-dimensional PCA score plot (Fig. 1B). PCA analysis indicated that metabolic alterations did occur in the serum of AAV patients with renal involvement. The variable importance in the project (VIP) was calculated using OPLS-DA to further demonstrate the differences between the two groups. The model had an R2Y value of 0.966, which meant that it explained 96.6% of the variance observed within the data, and a Q2Y value of 0.886, showing that the model was highly predictive. The OPLS-DA score plot also showed a significant difference between the disease and control groups (Fig. 1C). Then we performed 200 permutation verification experiments on this OPLS-DA model, R2′ and Q2′ were found to be smaller than R2 and Q2 of the original model (Fig. 1D), indicating that the model was meaningful and could be used to screen for differential metabolites based on the VIP values.

## Differentially expressed metabolites were statistically characterized to capture metabolic changes

We obtained VIP values from the OPLS-DA analysis and calculated fold change values for each metabolite. Metabolites with VIP $\geq 1$ and metabolites with fold change $\geq 2$ or $\leq 0.5$ were generally considered significant. Based on this criterion, 135 DEMs were selected, with 121 upregulated and 14 downregulated. The DEMs mainly consisted of amino acid, nucleotide, organic acid, bile acids eicosanoid, and their derivatives. The results were visualized in a volcano plot as shown in Fig. 2A and the DEMs were clustered and shown by a heatmap diagram in Fig. 2B. The top 10 upregulated DEMs and the top 10 downregulated DEMs were displayed in Fig. 2C and the top 20 DEMs ranked according to VIP values were presented in Fig. 2D.

## Characterizing altered metabolic pathway in AAV patients with renal involvement by KEGG annotation and metabolic set enrichment analysis

The KEGG database was used to annotate pathways for DEMs, and 92 pathways were engaged, mostly by amino acid metabolism and nucleotide metabolism. We then used metabolic set enrichment analysis to identify metabolic pathway sets with distinct biological functions. As shown in Fig. 3, pyrimidine metabolism, cysteine and methionine metabolism, tryptophan metabolism, glyoxylate and dicarboxylate metabolism, and D-glutamine and D-glutamate metabolism were significantly enriched.

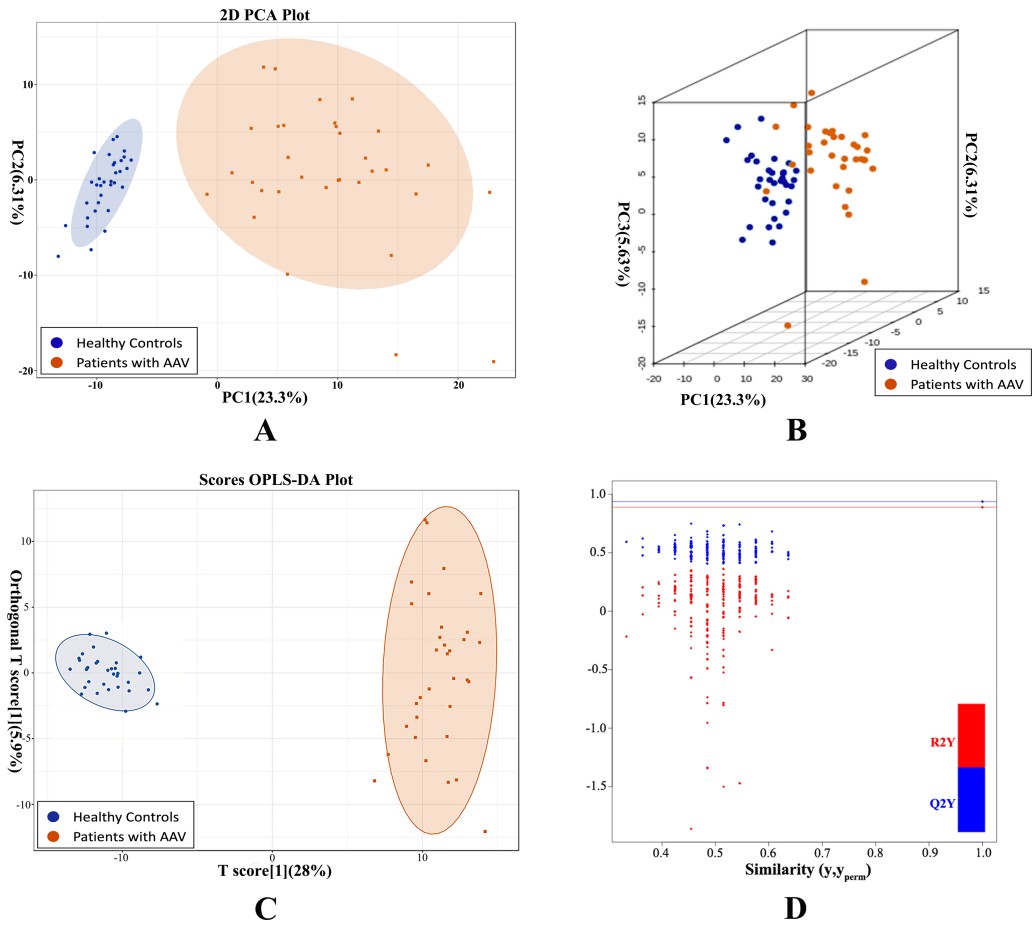

**Figure 1 PCA and OPLS-DA demonstrated a significant metabolic difference between patients with AAV and healthy controls.** (A) Plane score plot of the PCA analysis; (B) 3D score plot of PCA analysis; (C) OPLS-DA score plot of OPLS-DA model; (D) permutation test of OPLS-DA model.

## Identifying DEMs that could accurately distinguish AAV patients with renal involvement from healthy controls

ROC analyses of individual metabolites were performed to verify plasma metabolites with high selectivity and specificity in identifying AAV patients with renal involvement. N-acetyl-L-leucine, Acetyl-DL-Valine, and 5-hydroxyindole-3-acetic acid exhibited remarkable diagnostic capacity with very high AUC values of 1, higher than 0.987 for creatinine (Fig. 4A). The significantly different expression levels of these DEMs and creatinine between patients and healthy controls were visualized as violin plots in Fig. 4B. Furthermore, subjects were divided into four combinations by disease and health, male and female, and each combination was randomly split into training and validation sets in a 2:1 ratio. LASSO, XGBoost, and logistic regression were combined to calculate the best diagnostic regression model in the training set. As shown in Table 2, the optimal logistic regression model (AIC = 6.00) was formed from two candidate biomarkers, 1-methylhistidine and Asp-phe. The model was evaluated with the training and validation sets separately, and both showed

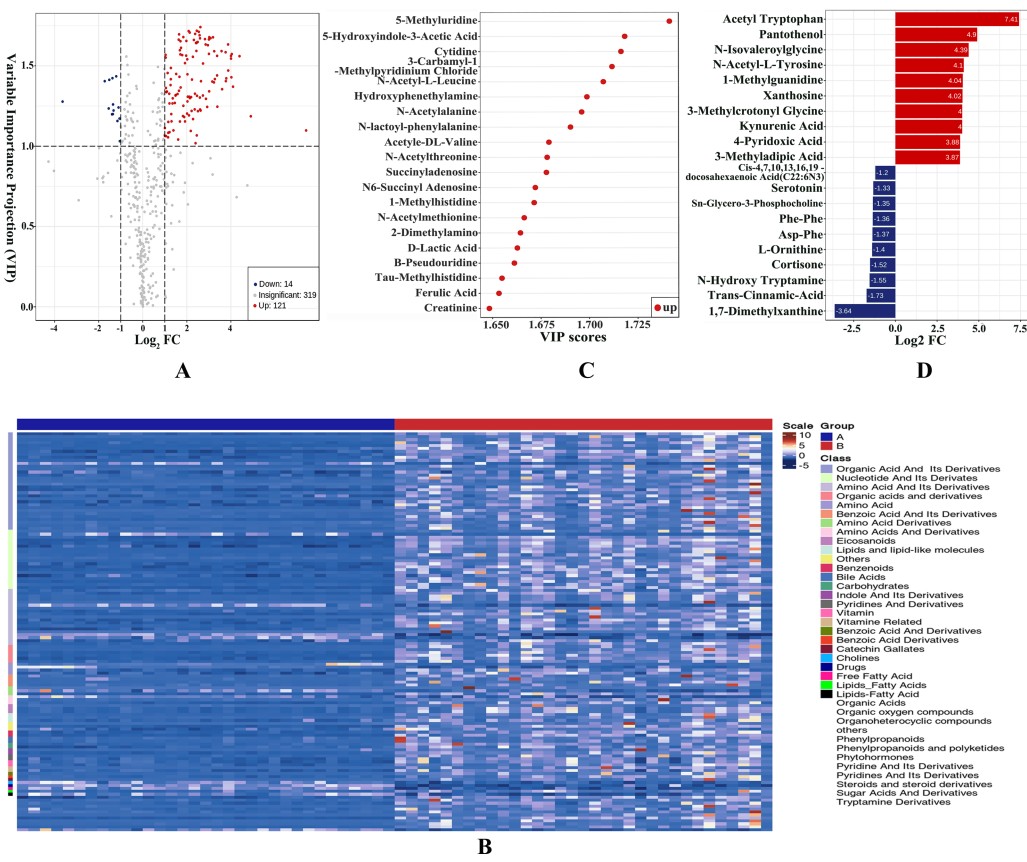

**Figure 2** Differentially expressed metabolites (DEMs) were statistically characterized under VIP and fold change double screening and visualized to capture metabolic changes. (A) Volcano plot under VIP + Fold Change double screening condition; (B) heatmap overview of all DEMs and samples clustered by cluster analysis; (C) bar plots of top 10 upregulated DEMs and top 10 downregulated DEMs; (D) the top 20 DEMs with the largest VIP values in OPLS-DA model.

extremely high sensitivity and specificity in diagnosis with an AUC value of 1 (Fig. 5A). And these two metabolites differed significantly between the AAV renal involvement group and the healthy control group in both training and validation datasets (Fig. 5C).

## Screening DEMs associated with the progression and prognosis of AAV patients with renal involvement

We set the endpoint event as entry to end-stage renal disease or death and followed patients until December 2019. AAV patients with renal involvement were divided into two groups (the events group and no-events group) based on the occurrence of endpoint events at the end of follow-up. An independent sample $T$-test was performed and we found that 1-methylhistidine, N-acetyl-L-leucine, 2-dimethylamino guanosine, N-acetylalanine, cytidine, and adenosine O-ribose were expressed differently between the events group and the no-events group, while ANCA showed no statistical difference between the two groups (Fig. 6A). Spearman correlation coefficient analysis among DEMs selected above, BVAS scores, ANCA, age, gender, and clinical characteristics reflecting the degree of renal

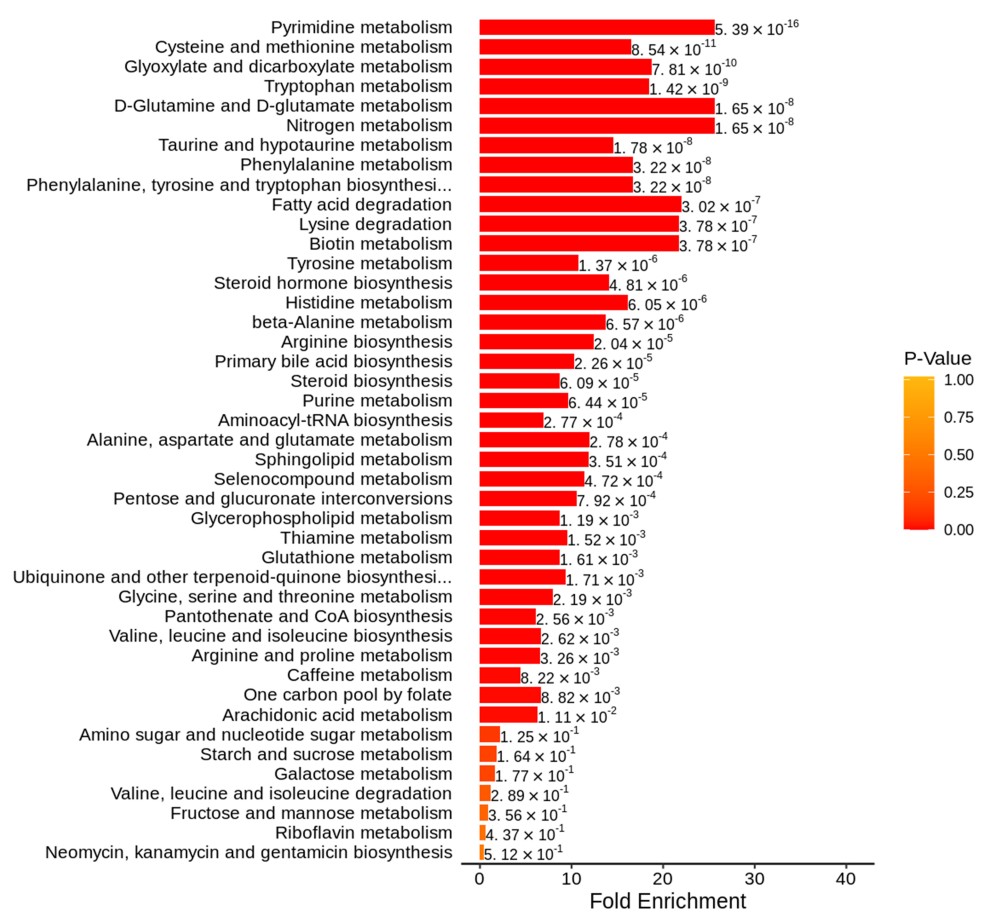

**Figure 3** **Metabolite set enrichment analysis diagram.** The metabolite sets with top 50 *P*-values were displayed.

injury, including eGFR, creatinine, and BUN (Table 3). ANCA showed no correlation with gender, age, creatinine, BUN, eGFR and BVAS, which indicated that ANCA cannot assess disease progression. All the selected DEMs exhibited no statistically significant difference in age and gender, so the interferences of age or gender could be eliminated. All the selected DEMs were significantly related to creatinine, which might result from the accumulation of metabolites due to impaired renal function. However, these DEMs are also related to BVAS, so AAV also plays an important role in their metabolic changes. To draw Kaplan–Meier survival curves with end-point events, patients were divided into two groups according to their creatinine levels, and those whose creatinine levels were higher than 442, indicating that they entered the stage of renal failure, were in group 2 (Fig. 6B). The renal survival time of patients with high creatinine was significantly shorter than that of patients with low creatinine ($P = 0.0015$), consistent with the general consensus. We also divided patients into two groups based on the 1-methylhistidine median level of 1.55, and made Kaplan–Meier survival curves (Fig. 6C). Patients with plasma 1-methylhistidine levels higher than 1.55 had significantly shorter renal survival times than patients with low

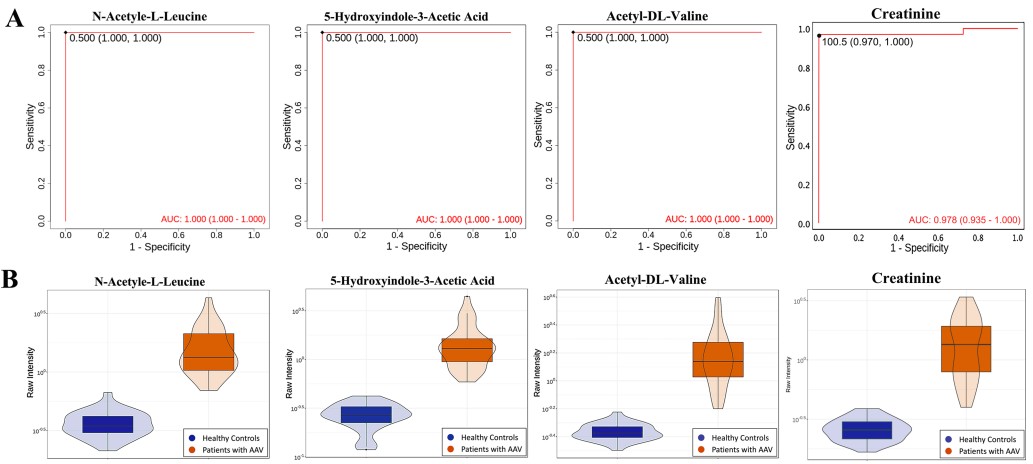

**Figure 4** **N-acetyl-L-leucine, Acetyl-DL-Valine, and 5-hydroxyindole-3-acetic acid could accurately distinguish AAV patients with renal involvement from healthy controls.** (A) ROC curves of N-acetyl-L-leucine, Acetyl-DL-Valine, 5-hydroxyindole-3-acetic acid and creatinine; (B) violin plots of N-acetyl-L-leucine, Acetyl-DL-Valine, 5-hydroxyindole-3-acetic acid, and creatinine.

**Table 2** **The best logistic regression model.**

| Metabolite name | Coefficient |
| --- | --- |
| Intercept | 88.08 |
| 1-Methylhistidine | 213.83 |
| Asp-Phe | −187.56 |

1-methylhistidine levels ($P = 0.046$). 1-methylhistidine was significantly associated with the progression and prognosis in patients with AAV-associated renal impairment, and further study of its role in this disease may contribute to the discovery of new biomarkers or therapeutic targets.

## DISCUSSION

Renal damage is one of the main causes of death in AAV patients, and its prognosis is closely related to the patient's renal function at the time of diagnosis. However, patients are often diagnosed in the stage of renal failure due to the insidious onset and rapid progression. As an important serum biomarker for the diagnosis and treatment of AAV, the role of ANCA in assessing disease activity and prognosis prediction remains controversial (*Finkielman et al., 2007*; *Tomasson et al., 2012*). Biomarkers that can monitor the progression and predict the prognosis of AAV with renal involvement are urgently needed to improve patients' therapeutic outcomes.

With the help of metabolomics analysis, we obtained information on changes in serum metabolites and related metabolic pathways in patients with AAV renal impairment. In this study, we detected 455 metabolites based on broadly targeted metabolomic techniques and successfully identified metabolic differences between AAV with renal involvement groups

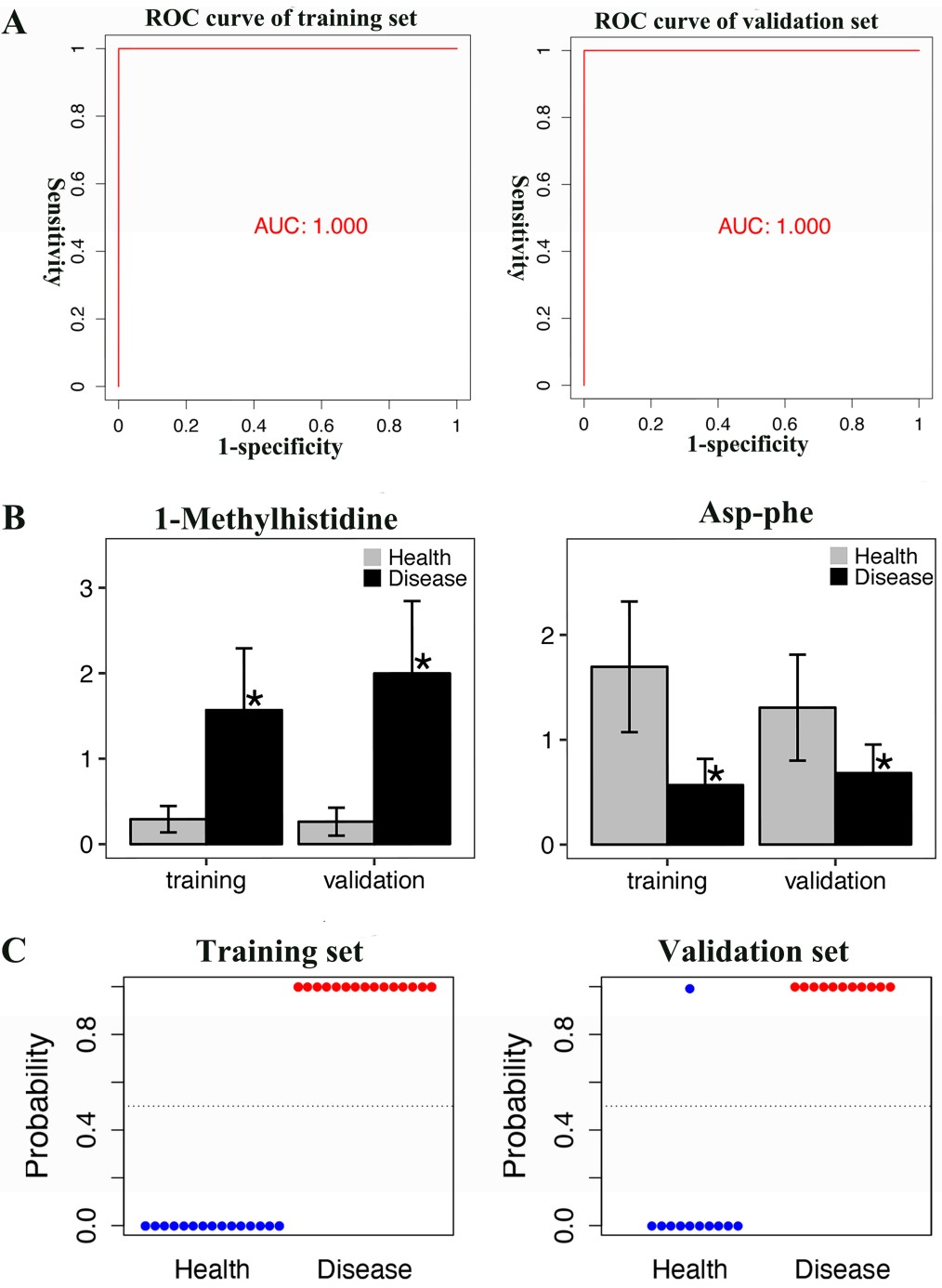

**Figure 5** **The best regression model calculated by LASSO, XGBoost and logistic regression analysis was evaluated in both training and validation sets, and showed extremely high sensitivity and specificity in diagnosis.** (A) ROC curve of training set and validation set; (B) $T$-test histogram of 1-Methylhistidine and Asp-Phe between health group and disease group in training set and validation set, an asterisk (*) means $p < 0.05$; (C) Scatter plots of the predicted results of the logistic regression model in training set and validation set.

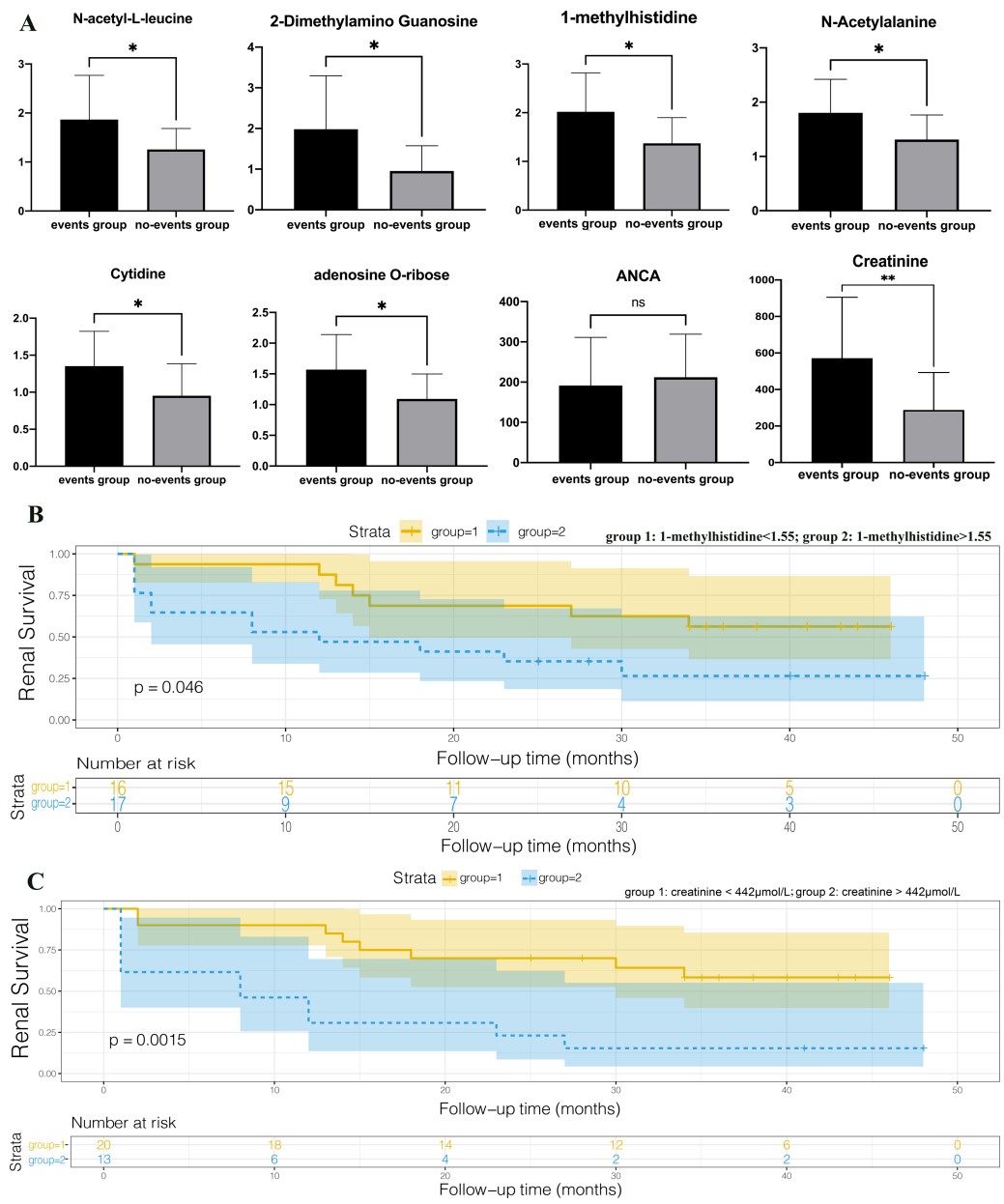

**Figure 6** **1-methylhistidine associated with the progression and prognosis of AAV patients with renal involvement.** (A) *T*-test histograms of 1-methylhistidine, N-acetyl-L-leucine, 2-dimethylamino guanosine, N-acetylalanine, cytidine, adenosine O-ribose, ANCA and creatinine between events group and no-events group. The expression of the six metabolites was significantly up-regulated in the end-point event group, while ANCA showed no statistical difference between the two groups. * *p* < 0.05, and ns means no statistical difference; (B) Kaplan–Meier survival curves of the two groups with high and low plasma creatinine; (C) Kaplan–Meier survival curves of the two groups with high and low plasma 1-methylhistidine.

**Table 3  The statistics of spearman correlation coefficient among metabolites, ANCA, and clinical characters.**

| $r_s$ | Gender | Age | creatinine | BUN | eGFR | BVAS |
|---|---|---|---|---|---|---|
| ANCA | 0.058 | −0.081 | −0.188 | 0.224 | −0.225 | −0.344 |
| 1-methylhistidine | −0.146 | 0.244 | 0.597[**] | 0.496[**] | −0.519[**] | 0.564[**] |
| N-acetyl-L-leucine | −0.064 | −0.055 | 0.606[**] | 0.369[*] | −0.379[*] | 0.375[*] |
| N-acetylalanine | 0.07 | 0.135 | 0.595[**] | 0.312 | −0.369[*] | 0.576[**] |
| adenosine O-ribose | 0.032 | 0.21 | 0.570[**] | 0.393[*] | −0.438[*] | 0.454[**] |
| 2-dimethylamino guanosine | 0.115 | 0.118 | 0.634[**] | 0.209 | −0.282 | 0.495[**] |
| cytidine | −0.025 | 0.276 | 0.548[**] | 0.193 | −0.208 | 0.465[**] |

**Notes.**

$r_s$, Spearman correlation coefficient.

[*] $p < 0.05$.

[**] $p < 0.01$.

and healthy controls by PCA and OPLS-DA analysis. 135 metabolites were identified as the DEMs in AAV with renal involvement groups, which were involved in 92 altered metabolic pathways.

Based on clinical data, we identified some metabolites that could accurately distinguish patients with AAV renal impairment from healthy controls in this study, as well as metabolites that were significantly associated with disease progression and prognosis. ROC curve analyses revealed that N-acetyl-L-leucine, Acetyl-DL-Valine, 5-hydroxyindole-3-acetic acid and the combination of 1-methylhistidine and Asp-phe have the highest sensitivity and specificity to distinguish patients with AAV renal impairment from healthy controls. These metabolites have the potential to be new diagnostic markers and need to be verified by further studies. N-acetyl-L-leucine is a derivate of the essential amino acid leucine, and it is often used to treat vestibular diseases and improve ataxia as a drug that could regulate vestibular function (*Günther et al., 2015*; *Tighilet et al., 2015*). However, its role in AAV with renal involvement or autoimmune disease has not been reported. N-acetyl-L-leucine and acetyl-DL-valine are derivates of the branched-chain amino acid leucine and valine; their upregulation in AAV with renal involvement might reflect the upregulation of branched-chain amino acid. Previous study of branched-chain amino acid suggest that high concentrations of branched-chain amino acid can damage circulating blood cells and contribute to the pro-inflammatory and oxidative status observed in several pathophysiological conditions (*Zhenyukh et al., 2017*). Asp-Phe is also a derivative of an amino acid. From the result of KEGG annotation in this study, we can find that amino acid metabolism is highly positive in the AAV with renal involvement group, which might be related to the effects of amino acids in promoting protein synthesis and lymphocyte proliferation during the active phase of vasculitis (*Coras, Murillo-Saich & Guma, 2020*).

1-methylhistidine was found to be significantly associated with the progression and prognosis of AAV patients with renal involvement. 1-methylhistidine significantly increased in patients with prognoses of end-stage renal disease or death and was positively related to the renal survival times of patients. 1-methylhistidine is a metabolic byproduct of anserine (beta-alanyl-L-1-methyl-histidine), a carnosine analog (*Hu et al., 2019*). Carnosine and its analog have been recognized to play a powerfully protective role in oxidative and
nitrosative stress and have the potential to inhibit multiple mechanisms of injury after hypoxia–ischemia (*Bellia et al., 2011*). Oxidative and nitrosative stress and hypoxia-ischemia injury are key links in the development of AAV with renal involvement, so the significant increase of 1-methylhistidine might indicate that carnosine and its analog participate in antagonizing AAV with renal involvement. Whether 1-methylhistidine has the value of being a prognostic biomarker and the role of its related metabolic pathway changes in AAV renal damage deserves further study.

KEGG annotation and metabolite set enrichment analysis demonstrate that amino acid metabolism, including cysteine and methionine metabolism, tryptophan metabolism, and D-glutamine and D-glutamate metabolism, change tremendously in the AAV with renal involvement group. It is acknowledged that amino acid catabolism is an important node in controlling immune response (*Grohmann & Bronte, 2010*; *Murray, 2016*). Cysteine and methionine metabolism and D-glutamine and D-glutamate metabolism are associated with oxidative stress, inflammation, and specific immunity (*Go & Jones, 2011*; *Jain et al., 2009*; *Wang & Green, 2012*). The metabolism of tryptophan has also been linked to inflammatory reactions and immune regulation (*Günther et al., 2020*). In this study, kynurenic acid and kynurenine upregulated significantly while serotonin and N-hydroxy tryptamine showed a significant downregulation, which indicated the activation of the tryptophan-kynurenine pathway in AAV patients with renal involvement. Some studies suggest that the tryptophan-kynurenine pathway plays a protective role by counter regulating the immune response during inflammation (*Bauer et al., 2005*; *Günther et al., 2020*; *Wang et al., 2006*), while other research shows that the tryptophan-kynurenine pathway could promote the renal damage progression in AAV (*Barth et al., 2009*). Therefore, the activation of the tryptophan-kynurenine pathway is a key link in the development of renal damage in AAV. Investigating the mechanisms of the tryptophan-kynurenine pathway in AAV with renal involvement may facilitate the discovery of therapeutic targets and improve the therapeutic outcomes of AAV patients with renal involvement.

This study has several constraints. Although we included all untreated patients with a first diagnosis of AAV, the majority of patients in this study had renal insufficiency due to the insidious onset and rapid progression of AAV. Metabolic changes in this study were the result of the combined action of AAV and renal insufficiency, and the accumulation of metabolites caused by renal insufficiency had a great influence on the outcome. Due to the absence of two controls, patients with AAV but without renal impairment and patients with renal impairment but without AAV, we could not distinguish between metabolic changes caused by renal insufficiency and those caused by AAV. Therefore, the results of this study are only applicable to the cases of AAV with renal impairment. But we believe that our results still have some reference value for researchers who want to conduct AAV related research. Firstly, a recent study used metabolomics analysis to investigate the metabolic differences between the active and the remission phase of 10 AAV patients with renal impairment. They found that amino acid metabolism and nucleotide synthesis were significantly higher in the active phase samples, which was consistent with our results (*Geetha et al., 2022*). Secondly, our results showed that the major metabolic change in the AAV patients with kidney damage was in amino acid metabolism pathway. The essential

amino acids and their metabolites elevated significantly or not significant altered in the AAV patients in our study. However, it is widely acknowledged that plasma essential AAs (EAAs), notably branched-chain AAs (BCAAs), decrease in patients with chronic renal failure (*Canepa et al., 2002*; *Divino Filho et al., 1997*; *Suvanapha et al., 1991*). Laidlaw's study found that valine, tyrosine, arginine, serine, BCAA, and total essential amino acids significantly decreased in renal failure patients than healthy control (*Laidlaw et al., 1994*). A new study showed that plasma concentrations of lysine, methionine, threonine, tryptophan, valine, alanine, asparagine, glutamine, serine, and tyrosine were all lower in renal failure patients before hemodialysis compared to controls (*Post et al., 2022*). Therefore, we think that the changes in amino acid metabolism in this study are more related to AAV. Thirdly, there were 11 differential metabolites (no essential amino acids) reducing in AAV patients with renal involvement in our study, which could not be attributed to the accumulation of metabolites resulted from impaired renal function and was likely to be associated with AAV. Collectively, we believe that AAV was the key factor of metabolic change in this study. We will include patients with AAV but no renal impairment and patients with renal impairment but no AAV to distinguish the metabolic changes caused by renal failure and AAV respectively, and further investigate the possible mechanism of metabolic changes in AAV patients with renal involvement in our future studies.

The second limitation was that despite two-years sample collection timespan, the sample size for biomarker screening was remained very limited because of the low prevalence of AAV. We are still collecting samples and will expand the sample size in our future studies to further verify our findings. Finally, this study was a single-center study, and the majority of the patients were from Hubei Province, China, therefore ethnic differences, diet and geographical factors may not have been avoided.

## CONCLUSIONS

Our metabolomic analysis of serum samples demonstrates that metabolic alterations do occur in AAV patients with renal damage. In this study, amino acid metabolism was found to be the most significantly altered metabolic pathway in AAV patients with renal impairment. We also identified some metabolites that could accurately distinguish patients with AAV renal impairment from healthy controls in this study, as well as metabolites that were significantly associated with disease progression and prognosis. Overall, this study provides information on changes in metabolites and metabolic pathways for future studies of AAV-related kidney damage and lays a foundation for the exploration of new biomarkers of AAV-related kidney damage.

### Funding

This work was supported by the National Natural Science Foundation of China (No. 81800609). The funders had no role in study design, data collection and analysis, decision to publish, or preparation of the manuscript.

## Grant Disclosures

The following grant information was disclosed by the authors:
National Natural Science Foundation of China: 81800609.

## Competing Interests

The authors declare there are no competing interests.

## Author Contributions

- Siyang Liu analyzed the data, prepared figures and/or tables, and approved the final draft.
- Qing Xu analyzed the data, prepared figures and/or tables, and approved the final draft.
- Yiru Wang analyzed the data, authored or reviewed drafts of the article, and approved the final draft.
- Yongman Lv conceived and designed the experiments, analyzed the data, authored or reviewed drafts of the article, and approved the final draft.
- Qing quan Liu conceived and designed the experiments, performed the experiments, authored or reviewed drafts of the article, and approved the final draft.

## Human Ethics

The following information was supplied relating to ethical approvals (i.e., approving body and any reference numbers):

This study was approved by the Medical Ethics Committee of Tongji Hospital of Huazhong University of Science and Technology (TJ-IRB20220159).

## Data Availability

The raw measurements are available in the Supplemental Files.

## Supplemental Information

Supplemental information for this article can be found online at http://dx.doi.org/10.7717/peerj.15051#supplemental-information.

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
