# Peer review of "Metabolomics combined with clinical analysis explores metabolic changes and potential serum metabolite biomarkers of antineutrophil cytoplasmic antibody-associated vasculitis with renal impairment"

_PeerJ, doi:10.7717/peerj.15051_

## Round 0.1 · original submission · Major Revisions

The manuscript has been evaluated by three experts in the field. They have highlighted several areas for major revision, with particular reference by more than one reviewer to the experimental design.

Reviewer 1 ·

Basic reporting

Thank you for asking me to review this manuscript.

This manuscript covers one of the unmet needs in AAV and application of metabolomics for biomarker and correlation of certain metabolites to predict clinical outcomes - ESRD in good hypothesis.

Use of basic English language remains very poor through out.e.g. statement in Intro- line 60 Renal is composed of a large number of small vessels and is rich in blood flow, so it is highly vulnerable to AAV. Even spelling mistakes are not corrected. e.g. Materials and methods sec 2.2 Sample preparation- Frozen samples were toked out.

Literature review and introduction is very poorly written. Many key studies in this area e.g. Tissue plasminogen antibody, LAMP-2 antibodies and use of biospectroscopy in AAV ( Hewins et al, Kain et al and Morris et al) were not discussed. Limitations of ANCA were not discussed.

There are factual errors like mentioning propylthiouracil as immunosuppressive drug.

Experimental design

Design has been flawed.
Authors have examined limited number of patients with AAV and healthy controls. They don't mention if samples were collected before commencing any immunosuppression. Drug details and ANCA details are not provided in raw data. For 2 years study period and further 2 years of observations- paired samples of AAV patients in active and remission state would have been better design.
Authors do not justify exclusion criteria- e.g. like T2DM.
Although they mention controls were age matched- average age data presented appears unmatched control.

Validity of the findings

The presenting concept of metabolomic signature of renal involvement in AAV was already proven with serology and histology. Metabolome described does help predict renal outcomes.

Role of metabolite- 1-methylhistadine in predicting renal prognosis is interesting and novel. Further correlations with ANCA risk scoring equation would have been helpful to compare predictive value of metabolomics, given small sample size and number of events (Brix et al KI 2018)

Additional comments

Authors fail to acknowledge any limitations. Their conclusion is rather strongly worded for small sample size. Referencing is difficult to follow and Figure legends are difficult to read.

Overall very poorly conducted and presented study.

If this is to be considered again then at least they should perform paired (active Vs Remission) analysis, consider ANCA negative patients. Needs attention to basic English

Reviewer 2 ·

Basic reporting

Article is professionally written in clear and concise english. I however, found it difficult to follow the plots and charts generated in the paper to have small fonts and extremely difficult to read. I hope, authors improve the quality of plots that they have generated. I would like to specifically improve the size of fonts so that the plots can be read without zooming in.

Experimental design

This article, falls within the scope of the journal and authors provide a robust explanation of the data used for the study and their interpretations of it.

Validity of the findings

Article clearly makes sound and statistically relevant conclusions and is fit for publication in the journal

·

Basic reporting

Thanks for allowing me to review this interesting paper.

STRENGTHS
The question that this study attempts to address is a very important one. We do need biomarkers that are better able to report ANCA vasculitis disease activity.

The paper is well-structured and the figures and tables well-constructed and informative.


LIMITATIONS
1) There are quite a few errors in syntax / grammar that make the paper a bit difficult to read (e.g. "Renal is composed of a large number of small blood vessels and is rich in blood flow, so it is highly vulnerable to AAV.")

2) Unless I have missed it, the metabolomics data are not deposited in any publicly accessible database. This would be desirable - and in fact I believe that PeerJ mandate this.

Experimental design

STRENGTHS
The metabolomic profiling appears to be comprehensive and the analysis is sensible and unbiased.


LIMITATIONS
MAJOR
1) The major limitation is that there are two critical control groups missing: patients with AAV but no renal impairment and patients with renal impairment but no AAV. Without these additional control groups, it is impossible to differentiate between metabolic changes that are associated with kidney impairment and those that are associated with AAV. (And I think the most likely assumption is that the vast majority of changes reported are likely to be associated with the renal failure rather than AAV per se.)

2) The ethical basis for this study is potentially controversial. The participants did not provide informed consent for their samples to be used for research purposes.


MINOR
3) There is no attempt to justify sample size. This cohort seems quite small for for biomarker discovery - especially when splitting into training and validation sets. Can the authors demonstrate that this study was adequately powered?

4) There were a number of exclusion criteria. Why were these chosen?

5) Were the storage conditions / delays in sample processing the same in the AAV and healthy control groups?

Validity of the findings

The main results are valid but some of the conclusions are not warranted, because of the lack of relevant control groups.

For example, "Sensitive biomarkers for early diagnosis and promising biomarkers for progression monitoring and prognosis predicting of AAV are identified successfully." The population study here all had disease that was not in its early stages (because they had significant renal impairment) and so it is not possible to test whether biomarkers perform well for early diagnosis.

For example, "This study identified metabolomic differences between AAV patients with renal involvement and non-AAV individuals, and the LC-MS/MS based serum metabolomics
approach had the potential ability to become an objective diagnostic method for
diagnosing AAV." If the metabolomic differences were primarily due to differences in kidney function between the disease and control groups (which seems likely), then they would not have any potential use in diagnosing AAV.

---

## Round 0.2 · Minor Revisions

The manuscript is clearly improved in many aspects, however, the reviewers have both highlighted several points where clarity is needed. Specifically, details on when samples were collected are crucial and recognition of the limitation of the lack of additional control groups. Reviewer 3 has highlighted several points on the validity of the findings, where some relatively simple sensitivity analyses may help to clarify these data better. We look forward to receiving your corrected manuscript.

Reviewer 1 ·

Basic reporting

There has been significant improvement in presentation

Experimental design

Again there are limitations in the design. While authors have expanded on exclusion criteria-all of the criteria are not well justified. e.g. other renal disease - it is important to show that the metabolomic signature is unique to AAV from other immune mediated renal conditions making it a potential biomarker. Future plans of collecting more samples/paired samples is not really rebuttal for this piece of research.

Validity of the findings

This study provides clear new findings and have potential for translational research.

Additional comments

There are still minor grammatical errors - e.g. in section 2.2 'Equal volume of separate samples- this should be separated samples.

·

Basic reporting

Thank you very much for considering my comments during the first round of review. I am satisfied that you have addressed most of these.

There are still some things that I think are a little unclear.

1) Could you please expand on exactly when these samples were collected, relative to disease timecourse. Were these all taken at the time of first disease presentation? Were they collected before or after treatment was initiated? (This is essential to know when interpreting your results, because if the samples were taken after treatment was initiated then the metabolomic changes may reflect treatment effects - e.g. from high dose glucocorticoids - rather than disease-specific effects.). I think these are all pre-treatment samples as the methods state that you "excluded... patients taking immunosuppressive drugs"? But it would be helpful to state that explicitly please in the methods.

Experimental design

2) The authors have addressed my initial concerns. I still think it would be desirable to include additional control groups (renal failure without vasculitis and vasculitis without renal failure) - but I appreciate that this would constitute a major body of additional work and may be better submitted as a future publication. However, if the authors wish to proceed without these additional groups, they should make sure to acknowledge this major limitation (see below).

Validity of the findings

3) I do still think that it is very hard to know how to interpret the findings without the additional control groups mentioned above. Serum creatinine was significantly correlated with all of the metabolites presented in table 3, which the authors acknowledge may be because these metabolites are accumulating because of renal impairment (rather than ANCA vasculitis per se) - lines 283 and 284. So I think it is important that this study is presented as an exploratory, hypothesis-generating analysis. It is useful in exploring the potential utility of metabolite-based biomarkers in ANCA vasculitis but cannot yield any validated biomarkers because of this limitiation.

Therefore, I wonder if the work presented in results sections 3.5 and 3.6 (screening for diagnostic and prognostic biomarkers respectively) is in danger of becoming invalid. As a sensitivity analysis, it might be helpful to also do the ROC curve for serum creatinine (in Fig 4) and the boxplot / Kaplan-Meier analysis for serum creatinine (Fig 6). I suspect these would also show highly significant results and, if included in these figures, would serve to remind the reader that this study cannot differentiate between the effects of vasculitis and the effects of renal impairment. If, on the other hand, serum creatinine performs poorly in these analyses, then that would be an argument that these metabolomic changes are more likely to be due to AAV than to renal failure.

The authors do state in closing that, "Secondly, although we included all untreated patients with a first
diagnosis of AAV, the majority of patients in this study had renal insufficiency due to the insidious
onset and rapid progression of AAV. Metabolic changes might be the result of the combined action
of AAV and renal insufficiency, thus, the findings of this study are only applicable to the cases of AAV with renal involvement." This is helpful, but I don't think it is a strong enough statement. It may be that the metabolomic changes observed in this study are predominantly due to renal insufficiency. The conclusions in the abstract are also not fully supported by the data. A more measured conclusion may be more appropriate (e.g. that there are striking metabolomic changes in AAV patients with renal failure and that these might form the basis of prognostic biomarkers, but that additional studies would be required to determine that). (The abstract differs slightly the two times it appears in the combined .pdf so I am not sure which the intended version is - but either way I don't think this study can claim to have identified diagnostic or prognostic biomarkers.)

---

## Round 0.3 · accepted · Accept

Having reviewed the amended manuscript, I am satisfied that all points have been addressed adequately.